# Structural puzzles in virology solved with an overarching icosahedral design principle

Reidun Twarock [ID] [1]* & Antoni Luque [ID] [2]*

Viruses have evolved protein containers with a wide spectrum of icosahedral architectures to protect their genetic material. The geometric constraints defining these container designs, and their implications for viral evolution, are open problems in virology. The principle of quasi-equivalence is currently used to predict virus architecture, but improved imaging techniques have revealed increasing numbers of viral outliers. We show that this theory is a special case of an overarching design principle for icosahedral, as well as octahedral, architectures that can be formulated in terms of the Archimedean lattices and their duals. These surface structures encompass different blueprints for capsids with the same number of structural proteins, as well as for capsid architectures formed from a combination of minor and major capsid proteins, and are recurrent within viral lineages. They also apply to other icosahedral structures in nature, and offer alternative designs for man-made materials and nanocontainers in bionanotechnology.

[1] Departments of Mathematics and Biology, York Cross-disciplinary Centre for Systems Analysis, University of York, York YO10 5GE, UK. [2] Department of Mathematics and Statistics, Viral Information Institute, and Computational Science Research Center, San Diego State University, 5500 Campanile Drive, San Diego, CA 92182-7720, USA. *email: reidun.twarock@york.ac.uk; aluque@sdsu.edu

Polyhedral designs are ubiquitous in nature. They are fundamental for our understanding of molecular architectures in chemistry and physics[1], and occur at different length scales, from marine organisms[2] to protein nanocontainers with different biological functions[3,4]. Prominent examples are viruses, the most abundant biological entities on the planet[5] and the causative agents of some of the most devastating diseases known. Viruses store and protect their genetic material in protein containers called capsids[6], that vary in size and structural complexity. They range from 20 nm to 800 nm and consist of only a few dozen to thousands of coat proteins (CPs). The majority of viruses adopt polyhedral designs with icosahedral symmetry[7,8], that is, their CP positions conform to polyhedral blueprints that exhibit the characteristic arrangement of the rotational symmetry axes of an icosahedron (Fig. 1a).

Viruses exhibit this high degree of symmetry as a consequence of a principle that Crick and Watson termed genetic economy, namely, the limited capacity in the viral genome to code for the CPs forming its surrounding capsid[9]. This favours such symmetric architectures, because icosahedral symmetry has 60 different symmetry operations[10], reducing the cost of coding for the capsid by 1/60th, whilst creating a container with sufficient volume to store the viral genetic material. Caspar and Klug extended this idea by introducing the principle of quasi-equivalence[11], which explains how proteins can adopt locally equivalent, or quasiequivalent, positions in a capsid, by repeating this local configuration across the capsid surface. This allows larger viruses to form, requiring even smaller relative portions of their genomic sequences to code for their capsids, thus generating coding capacity for other viral components that are not present in smaller viruses and enabling more complex infection scenarios.

These two principles have dominated structural virology over the last 60 years. The infinite series of icosahedral blueprints introduced by Caspar and Klug is currently the major tool for the classification of virus structures[12]. However, increasing numbers of virus structures exhibit capsid protein numbers and layouts that fall outside the CK description, as discussed below for a wide range of examples. This indicates that there are fundamental design principles underpinning virus architecture, and implied geometric constraints on viral evolution, that are still not fully understood.

To address this, we revisit the construction of icosahedral architectures using the Archimedean lattices classified by Kepler in his classical Harmonices Mundi[13]. With these lattices, we are able to derive eight families of icosahedral polyhedra (derived from the lattices and their duals) that explain the outliers to the current classification scheme and at the same time provide an overarching design principle that encompasses the current models of virus architecture in Caspar-Klug theory. Using viruses from different families, we demonstrate that the icosahedral designs embodied by the polyhedral families derived here correspond to previously unsuspected capsid layouts in the virosphere and provide a different perspective on viral evolution. As we discuss below, this discovery also sheds new light on the many areas of science where icosahedral structures play an important role, and also provides designs for applications in bionanotechnology.

## Results

**Polyhedral models of icosahedral architecture.** Virus structures are prominent examples of icosahedral symmetry in biology. Their architectures are currently modelled and classified in terms of the series of Goldberg polyhedra[14]—three dimensional solids with pentagonal and hexagonal faces—that provide a reference frame for the positions of the capsid proteins (Fig. 1a). In particular, the polyhedral faces indicate the positions of pentagonal and hexagonal protein clusters called pentamers and hexamers, respectively. The same polyhedra also provide blueprints for the atomic positions of the fullerene cages in carbon chemistry, in particular the Buckminster fullerene known as the buckyball[1]. They also provide blueprints for the structural organisation of a wide range of both man-made and natural protein nanocontainers. Their duals, the geodesic polyhedra[15], are the architectural designs of the geodesic domes by Buckminster Fuller.

Goldberg polyhedra can be constructed from a hexagonal grid (lattice) by replacing 12 hexagons by pentagons (Fig. 1b), as required by Euler's Theorem to generate a closed polyhedral shape[16]. The distance $D$ between the pentagons at neighbouring fivefold vertices is the only degree of freedom in this construction, and can therefore be used to label the different geometric options in this infinite series of polyhedra. $D$ can only take on specific values that are constrained by the underlying hexagonal lattice geometry. In particular, using the hexagonal coordinates $h$ and $k$, which take on any integer values or zero to navigate between midpoints of neighbouring hexagons in the lattice, one obtains the following geometric restriction[11]:

$$T(h,k) := D^2(h,k)/A_0 = \left(h^2 + hk + k^2\right). \quad (1)$$

Here, $A_0$ corresponds to the area of the smallest triangle between any hexagonal midpoints, that is, the case $h = 1$ and $k = 0$—or equivalently, $h = 0$ and $k = 1$. A similar formula has been derived for elongated capsid structures[17].

$T$ is called the triangulation number (Fig. 1c) owing to its geometric interpretation in terms of the icosahedral triangulations obtained by connecting midpoints of neighbouring pentagons and hexagons, i.e., in terms of the dual (geodesic) polyhedra. $T$ indicates the numbers of triangular faces, called facets, in the triangulation that cover a triangular face of the icosahedron by area. The association of a protein subunit with each corner of such a triangular facet translates this infinite series of triangulations into the capsid layouts in quasiequivalence theory (Fig. 1d). Such blueprints only permit capsid layouts with $60T$ CPs, organised into 12 pentamers and $10(T-1)$ hexamers[11]. The condition expressed by Eq. 1 is therefore a geometric restriction on the possible values of $T$ and the possible CP numbers in the CK geometries. The initial elements of the series are $T = 1, 3, 4,$ and 7, and therefore the number of CPs contained in small icosahedral capsids are 60, 180, 240, and 420, respectively (Supplementary Table 1).

However, this is only one way in which an icosahedral structure can be built from repeats of the same (asymmetric) unit, and excludes geometries built from proteins of different sizes (such as a major and minor capsid protein) or capsids built from a protein in which one or several domains play distinguished roles. Such capsid layouts must be constructed from lattices in which every vertex is identical in terms of the lengths, numbers and relative angles of its protruding edges, but the relative angles between different edges at the same vertex can vary, reflecting occupation by different types of proteins or protein domains. From a geometric point of view, there are only 11 lattices (Chapter 2 in Grünbaum and Shephard[18]) that satisfy this generalised quasi-equivalence principle, which are the Archimedean lattices—also known as uniform lattices[13,16]. Among these lattices, only four contain a hexagonal sublattice (Fig. 2a). One of them is the hexagonal lattice itself on which the CK classification scheme is based. This lattice is labelled $(6, 6, 6)$ according to the types of regular polygons surrounding each vertex, in this case three hexagons. However, the hexagonal lattice is only the simplest grid that enables this construction. Other lattices containing hexagons at appropriate distances, that is, as a

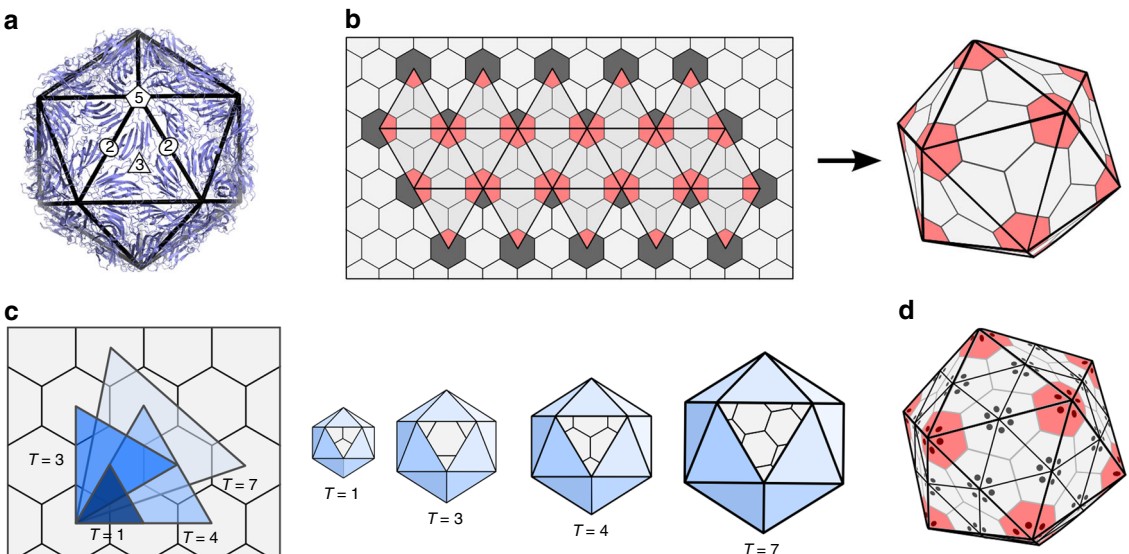

**Fig. 1** Capsid architecture according to Caspar and Klug theory. **a** Viruses exhibit the characteristic 5-, 3- and 2-fold rotational symmetry axes of icosahedral symmetry, indicated here with reference to the vertices, edges and faces of an icosahedral frame superimposed on the crystal structure of the $T = 1$ STNV shell (figure based on PDB-ID 2BUK). **b** Construction of an icosahedral polyhedron via replacement of hexagons in a hexagonal lattice by the equivalent of 12 equidistant pentagons (red). Dark grey areas indicate parts of hexagons in the lattice that do not form part of the surface lattice of the final polyhedral shape. **c** One of the 20 triangles of the icosahedral frame is shown superimposed on the hexagonal grid for the four smallest polyhedra that can be constructed in CK theory. These are shown in increasingly lighter shades of blue: $T(1, 0) = 1$, $T(1, 1) = 3$, $T(2, 0) = 4$ and $T(2, 1) = 7$. The corresponding polyhedra (right) have 20 identical triangular faces corresponding to the triangles (left), one of which is shown in each case. **d** The CP positions are indicated with reference to the dual polyhedron, that is, the triangulated structure obtained by connecting midpoints of adjacent hexagons and pentagons in the surface lattice. CPs are positioned in the corners of the triangular faces (shown here as dots), and result in clusters (capsomers) of six (hexamers) and five (pentamers) CPs. The example shown corresponds to a $T = 4$ layout, formed from 240 CPs that are organised as 12 pentamers (red) and 30 hexamers

hexagonal sublattice, are equally amenable to the CK construction, but have until now been ignored. These are the trihexagonal tiling $(3, 6, 3, 6)$, the snub hexagonal tiling $(3^4, 6)$, and the rhombitrihexagonal tiling $(3, 4, 6, 4)$ (Fig. 2a). These lattices are also called hexadeltille, snub hextille, and the truncated hexadeltille lattice, respectively[16].

By analogy to Caspar and Klug's construction, we classify the icosahedral polyhedra that can be constructed from these tilings via replacement of 12 hexagons by pentagons (Fig. 2b). Replacement of nearest neighbour hexagons results in each case in an icosahedrally symmetric Archimedean solid (Fig. 2c) that corresponds to the start of an infinite series of polyhedra, constructed by spacing the pentagonal insertions further apart. As a means to characterise different polyhedral structures in the series, we again use the hexagonal coordinates $h$ and $k$, now indicating steps between hexagonal midpoints in the hexagonal sublattice, to indicate the possible distances between the pentagonal insertions. In the three additional lattices, the midpoints of neighbouring hexagons are more distal than in the hexagonal lattice. Thus, the area covered by a triangular facet connecting midpoints of neighbouring hexagons (that is, the case $h = 0$ and $k = 1$, or vice versa) is larger than in the CK construction by a factor $\alpha_t = 4/3 \approx 1.33$ for the $(3, 6, 3, 6)$ lattice, $\alpha_s = 7/3 \approx 2.33$ for the $(3^4, 6)$ lattice, and $\alpha_r = 4/3 + 2/\sqrt{3} \approx 2.49$ for the $(3, 4, 6, 3)$ lattice, i.e., by factors corresponding to the relative sizes of the asymmetric lattice units (see coloured highlights in Fig. 2a). The $T$-number in the CK construction can therefore be scaled accordingly for the new lattices as follows

$$T_j(h, k) := \alpha_j\left(h^2 + hk + k^2\right) = \alpha_j \, T(h, k) , \qquad (2)$$

where $j = t, s, r$ indicates the lattice type used in the construction, denoting the trihexagonal, the snub hexagonal, and the

rhombitrihexagonal lattice, respectively. In particular, a polyhedron labelled $T_j(h, k)$ has the same number of pentagons and hexagons as a $T(h, k)$ Caspar Klug lattice, but the surface area covered by its faces is larger due to the additional polygons (triangles, squares) between the hexagons and pentagons. This is indicated by the scaling factor $\alpha_j$ that refers to the gain in surface area according to the planar lattice from which it is constructed as illustrated in Fig. 2.

The resulting geometries (Supplementary Tables 2–4) significantly widen the spectrum of possible icosahedral viral blueprints. For example, $T_t(1, 0) = 4/3$, $T_s(1, 0) = 7/3$ and $T_r(1, 0) = (4/3 + 2/\sqrt{3})$ are in between the $T(1, 0) = 1$ and $T(1, 1) = 3$ CK blueprints in terms of capsid size (Fig. 2d) if their hexagonal (sub)lattices are assumed to have the same footprint on the capsid surface, that is, same CP sizes. Additionally, some of these geometries constitute alternative layouts for similarly-sized CK geometries, such as $T_t(1, 1) = 4$ and $T_s(1, 1) = 7$ for $T(2, 0) = 4$ and $T(2, 1) = 7$ structures, respectively. In these cases, the alternative capsid models have the same relative surface areas, but are predicted to have different numbers and orientations of hexamers and pentamers, with interstitial spaces between these capsomers. These alternative structures (and their duals) correspond to previously unsuspected capsid layouts and offer a unifying framework for the classification of icosahedral virus architectures.

**Non-quasi-equivalent architectures in the HK97 lineage.** Increasing numbers of capsid architectures are reported with CP numbers and capsid layouts that are incompatible with the geometric blueprints of CK theory. Viruses with capsids formed from a combination of a major and minor capsid protein are examples that are challenging to interpret in the classical CK theory. Here

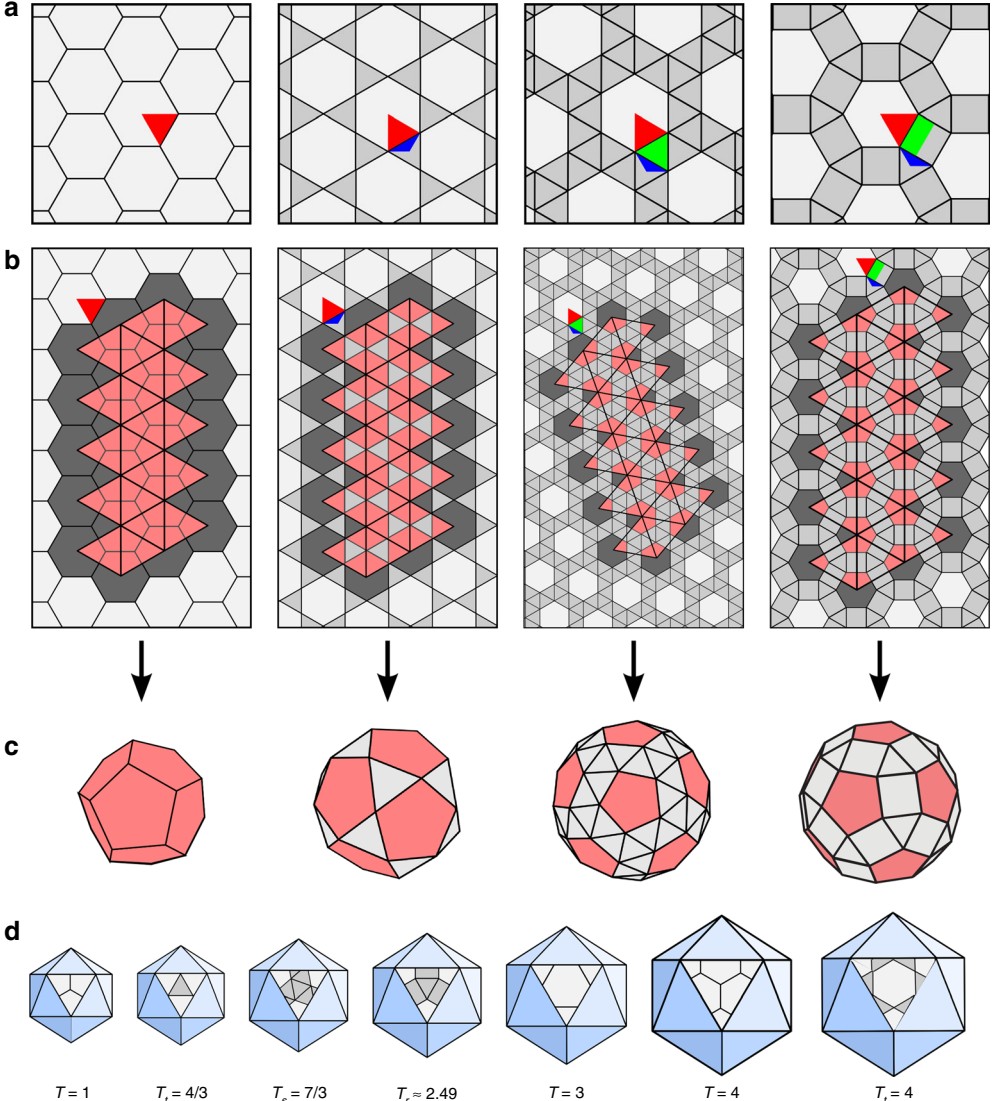

**Fig. 2** Design of icosahedral architectures from Archimedean lattices. **a** The four Archimedean lattices permitting the Caspar-Klug construction (from top to bottom): the hexagonal $(6, 6, 6)$, the trihexagonal $(3, 6, 3, 6)$, the snub hexagonal $(3^4, 6)$, and the rhombitrihexagonal $(3, 4, 6, 4)$ lattice. In each case, the asymmetric unit (repeat unit of the lattice) is highlighted. Its overlap with the hexagonal sublattice used for the construction of the icosahedral polyhedra is shown in red. Apart from the case of the hexagonal lattice, this also includes a third of a triangular surface (blue), and in addition a triangle or a half square (both shown in green) for two of the lattices, respectively. **b** Construction of Archimedean solids via replacement of 12 hexagons by pentagons in analogy to the Caspar-Klug construction (see also Fig. 1b). **c** The polyhedral shapes corresponding to the examples shown in **b**. They each correspond to the smallest polyhedron in an infinite series of polyhedra for the given lattice type. Folded structures for larger elements in the new series are provided in Supplementary Fig. 2. **d** The smallest polyhedral shapes ($T_t$, $T_s$ and $T_r$, denoting polyhedra derived from the trihexagonal, snub hexagonal and rhombitrihexagonal lattices, respectively) are shown organised according to their sizes in context with the Caspar-Klug polyhedra. As surface areas scale according to Eq. (2) with respect to the Caspar-Klug geometries, the new solutions fall into the size gaps in between polyhedra in the Caspar-Klug series, or provide alternative layouts for capsids of the same size, as is the case for $T(2, 0) = T_t(1, 1) = 4/3T(1, 1) = 4$

we provide examples from the HK97 lineage, demonstrating that such viruses can be rationalised in the Archimedian lattice framework proposed here.

The *Bacillus* phage Basilisk, for example, contains 1080 CPs, combining 540 major capsid proteins (MCPs) and 540 minor capsid proteins (mCPs)[19]. Using the relation $60\,T$ for CP numbers in CK theory, this would correspond to a $T$-number of 18, that is excluded by the geometric restriction in CK theory given by Eq. 1. If one only focuses on the 12 pentamers (more precisely, 11 pentamers and a putative portal) and 80 hexamers, then its structure would be classified as $T(3, 0) = 9$[19]. However, this ignores the 180 intersticial trimers and

misrepresents the relative orientations of the protein clusters as well as the surface area of the capsid (Fig. 3a). By contrast, Basilisk's CP positions are accurately represented by a $T_t(3, 0) = 12$ structure based on the trihexagonal lattice series in the framework of the overarching icosahedral design principle. This classification is also consistent with measurements of Basilisk's surface area ($1.69 \times 10^4$ nm$^2$, see Methods), that is comparable to the surface area of phage SIO-2 ($1.70 \times 10^4$ nm$^2$), which is a classical $T = 12$ capsid[20]. The Basilisk capsid is thus an icosahedral structure of similar size to that of a CK geometry, but exhibits a CP number and capsid layout that are not possible in the CK formalism.

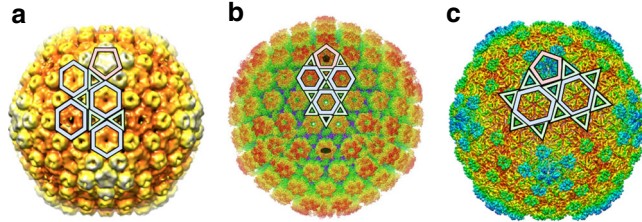

**Fig. 3** Viruses within a viral lineage adopting the same icosahedral series. Examples of viruses in the HK97 lineage, demonstrating that different members conform to the same family of icosahedral polyhedra: **a** Basilisk ($T_t(3, 0)$), **b** HSV-1 ($T_t(4, 0)$), **c** phage $\lambda$ ($T_t(2, 1)$). The building blocks of their polyhedral surface lattices are shown in red (pentagons), blue (hexagons), and green (triangles) superimposed on figures adapted from (**a**)[19], (**b**)[23] and (**c**)[25].

Basilisk (Fig. 3a) shares its MCP fold with other bacteriophages, archaeal and animal viruses in the HK97-lineage[12,21,22]. A reevaluation of other virus structures within this lineage reveals that these evolutionarily related viruses share the same underlying icosahedral lattice geometry, i.e., they belong to the same series of polyhedral designs (in this case, the trihexagonal series of $T_t$-architectures).

For example, herpes simplex virus type 1 (HSV-1) organises its MCP (VP5) in hexamers and pentamers with orientations reminiscent of those in the Basilisk capsid (Fig. 3b). The positions of these capsomers are consistent with the current classification of HSV-1 as $T(4, 0) = 16$. However, this misrepresents the relative orientations of the hexamers and ignores the secondary network of trimeric complexes between the capsomers that are formed from three mCPs (Tr1, Tr2a and Tr2b)[23]. The classification as a $T_t(4, 0) = 64/3$ structure in the new framework (Supplementary Table 2), however, accurately reflects both its 960 MCPs and 960 mCPs. The same holds for human cytomegalovirus (HCMV)[24] (structure not shown), which is structurally similar to HSV-1.

The mature capsid of phage $\lambda$ (Fig. 3c) is another example of a HK97-lineage virus with a trihexagonal icosahedral structure. It is currently classified as $T(2, 1) = 7$[12], but the orientation of the capsomers exhibits instead the layout of a $T_t(2, 1) = 28/3$ structure, because the protruding domains of the MCPs—rather than additional mCPs—occupy the triangular sublattice. These positions are also the locations of the reinforcement proteins gpD[25], highlighting the importance of these trimeric positions in the surface lattice (Fig. 3c). Alternatively, *Halorubrum sodomense* tailed virus 2 (HSTV-2), another member of the HK97-lineage, has been classified as $T(2, 1) = 7$. However, its capsid contains gpD-like trimers that occupy intersticial positions between capsomers, which is consistent with the trihexagonal structure $T_t(2, 1) = 28/3$ (see Fig. 8 in Pietilä et al.[26]). This implies an increase in capsid volume (and, consequently, genome size) by a factor of $\alpha_t^{3/2} \approx 1.54$ with respect to a classical $T(2, 1)$ capsid. This prediction is consistent with the empirical observation that HSTV-2 has a genome that is ~$1.4 - 1.7$ larger than that of $T = 7$ tailed phages[26], further corroborating its classification as a $T_t(2, 1) = 28/3$ capsid in our framework. Another example is the thermophilic bacteriophage P23-45, which is currently classed as a supersized $T = 7$ capsid architecture[27].

In summary, these examples suggest that the classification scheme for virus architecture introduced here highlights structural features shared by evolutionarily related viruses, and thus lends itself as a characteristic of viral lineages.

**Alternative capsid layouts with identical stoichiometry.** There are many examples of quasiequivalent viral capsids that are

formed from the same number of CPs, but exhibit different CP positions and capsomers. CK-theory does not distinguish between them. However, we demonstrate here based on the example of different $T = 3$ geometries, that the Archimedean lattices and their duals—called Laves lattices—provide a means to address this.

In CK theory, hexagonal surface lattices and their duals, corresponding to the triangular lattice $(3, 3, 3)$, are used interchangably. The smallest icosahedral polyhedron derived from a triangular lattice is the icosahedron, made of 20 triangles. The next largest is formed from 60 triangles, and provides a blueprint for a classical $T = 3$ structure. Using the convention of CK theory that polyhedral faces must represent groups of proteins that correspond, by number, to the rotational symmetry of the tile (e.g., triangles representing three proteins etc.), capsid layouts can be associated with polyhedral structures. Pariacoto virus (PAV; Fig. 4a), with its strong interaction between the three chains forming the triangular units, is an example of this type of $T^D(1, 1)$ surface architecture.

The duals of the other Archimedean lattices (trihexagonal, snub hexagonal, rhombitrihexagonal) present alternative surface architectures to those in CK theory in terms of rhomb, floret, and kite tiles, respectively (cf. Supplementary Table 5). Strictly applying the CK rule that the symmetry of a tile must be correlated with the number of proteins represented by the tile, singles out the dual trihexagonal lattices ($T_t^D$), i.e. the rhomb tilings with tiles representing clusters of two proteins (CP dimers). Rhomb tilings provide alternative layouts to the CK surface lattices, describing capsids with the same protein stoichiometry but different CP organisation. Bacteriophage MS2 (Fig. 4b), a virus assembled from 90 CP dimers, is an example of a $T = 3$ rhomb tiling ($T_t^D(1, 1)$; Supplementary Table 5). Note that whilst the protein stoichiometry in this case coincides with the CK framework, corresponding to the 180 proteins expected for a $T = 3$ structure, the identification as a $T_t^D(1, 1)$ geometry provides a more accurate account of CP positions and their relative orientations in the capsid surface.

**Non-quasi-equivalent and higher order rhomb tilings.** Extending the CK convention to allow rhombs to represent more than two CPs, as long as their positions on the tile respect the symmetry of the tile, higher numbers of proteins are also conceivable geometrically. This could be achieved, for example, by combining two dimers. The protein stoichiometry for such capsids would be 120 $T(h, k)$, and the first elements of the series would contain 120, 360 and 480 proteins. Picobirnavirus represents an example of the first element of this series (Supplementary Fig. 3a). This virus forms rhombus-like tiles made up of two protein dimers in parallel orientation, and contains 120 proteins in total[28]. This structure has been traditionally described as a forbidden $T = 2$ number in the CK framework, but it fits naturally into the new framework as a higher order rhomb tiling. The next elements of this series predict the existence of the forbidden numbers $T = 6$ (360 proteins) and 8 (480 proteins). Following this pattern, it is logical to think about the possibility of rhombus-like tiles representing three protein dimers, which would also satisfy the required twofold symmetry. The protein stoichiometry for these capsids would be 180 $T(h, k)$, and the three smallest geometries of this type would contain 180, 540 and 720 proteins. An example of the first element of this series is Zika virus (Supplementary Fig. 3b) in the *Flaviviridae* family. In particular, each rhomb tile in its capsid represents six elongated proteins (three dimers in parallel respecting the twofold symmetry of the tile), so that the 30 tiles represent 180 proteins in total. In pioneering work in 2002, the Rossmann lab and collaborators

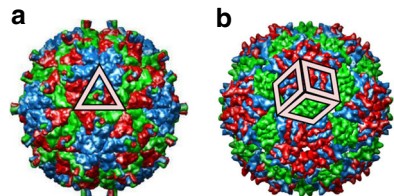

**Fig. 4** Capsid protein interfaces are constrained by icosahedral geometry. The classification of icosahedral designs distinguishes between capsid layouts of viruses formed from the same number of proteins. Examples of a triangle and rhomb tiling are shown: **a** Pariacoto virus ($T^D(1,1)$); **b** MS2 ($T_t^D(1,1)$). Tiles are shown superimposed on figures adapted from the ViPER data base (Pariacoto virus: PDB-id 1f8v[64]; MS2: PDB-id 2ms2[65])

realised that the three E monomers in each icosahedral asymmetric unit of Dengue virus[29] do not have quasiequivalent symmetric environments in the external, icosahedral scaffold formed from the 90 glycoprotein E dimers. Our approach based on the duals of the Archimedean lattices accommodates such non-quasiequivalent capsid structures.

Our framework thus extends the predictions of quasiequivalence theory by a more detailed understanding of capsid geometry, distinguishing between capsid architectures with different types of capsid protein organisation and interfaces given the same numbers of capsid proteins. This is important for a better understanding of the biophysical properties of viral capsids, such as their stability, and their roles in viral life cycles, e.g. during virion assembly and disassembly, and reveals geometric constraints on viral evolution.

## Discussion

These examples demonstrate that the overarching design principle for icosahedral architectures has been widely explored by nature, revealing an unsuspected spectrum of icosahedral capsid designs in the virosphere. This discovery opens up fundamental questions in virology.

The capsid architectures in CK-theory are the simplest possible icosahedral designs, realised by one type of CP that takes on different quasiequivalent positions in the capsid surface. The geometries described here, on the other hand, also include capsid layouts with two or more inequivalent geometric positions that are occupied either by a distinguished CP domain, or by a mixture of different CP types, e.g., MCP and mCP. Therefore, some of these capsid architectures incur a higher coding cost in term of genome length. The fact that nature realises these more complicated blueprints suggests that they must confer a selective advantage that is coupled to function. Such layouts may allow viruses to undergo conformational changes in their capsid structures[30], for example, through asymmetric components that brake the overall capsid symmetry[31], that enable more efficient genome release, or confer advantageous mechanical properties in terms of stability, stiffness and elasticity[32,33]. The mechanisms and pathways of capsid assembly are also likely to be different from the quasiequivalent capsid architectures in CK theory. For the latter, it is well understood how quasiequivalent conformations are defined via the tentacular interactions between CPs proposed by Harrison[34] based on the concept of tentacles introduced by Caspar[35]. The roles of viral genomes in the assembly of quasiequivalent capsid geometries are manifest in the packaging signal mediated assembly mechanism[36–40]. It is not clear, however, if the same principles apply to the more complex scenarios of the capsid architectures described here. Simulations of capsid assembly from triangular units reveal geometries that are akin to blueprints contained in two of the new series[41]. These simulations demonstrate that scaffold proteins are required for

the formation of these viral geometries, suggesting that additional components may be required for the assembly pathways associated with some of the viral blueprints introduced here. Moreover, the enhanced spectrum of viral designs unveiled here provides a different perspective on how viruses may have bridged the size gaps in their evolution of increasingly larger and complex capsid structures during evolutionary timescales.

Note that we have strictly adhered to the CK convention of representing capsid organisation by an edge-to-edge tiling in which the symmetry of every tile represents the numbers of proteins covered by this tile. We discuss here two ways in which predictive results can be achieved by relaxing any one of these conditions.

The first case involves the extension to non-edge-to-edge tilings. The protein counts for $T_t$ capsid architectures in Supplementary Table 2 are based on the relative sizes of the hexagonal and triangular faces of the lattice. The footprints of protein units occupying the hexagonal and pentagonal faces must be three times larger than those corresponding to the triangular faces, and such architectures are therefore either constructed from two types of proteins (an MCP and an mCP, with footprints in a ratio of 1:3), or a distinguished domain of the MCP occupies the smaller footprints of the triangular positions, taking on the role of the mCP. However, if a gyrated version of the trihexagonal lattice is used instead, in which the triangular face is rescaled such that its surface area is 3/5 of that of the pentagonal face (Fig. 5a), then a capsid blueprint is obtained in which all CP footprints are identical in size. An example of a virus following such a non-edge-to-edge tiling[18] is Pseudomonas phage phi6[42]. Its inner capsid is a pseudo $T = 2$ structure formed from 120 CPs, which is a CP number that is disallowed in CK theory, but rather follows the layout of a gyrated $T_t(1,0)$ lattice (Fig. 5b). The total number of CPs in such capsids corresponds to the sum of the protein counts indicated for MCP and mCP in Supplementary Table 2, i.e. to $n_p^{MCP} + n_p^{mCP}$.

The second case involves relaxing the symmetry condition on tiles. In the Results section, we have strictly adhered to the CK convention that the CP number represented by a tile must correspond to its rotational symmetry. By relaxing this requirement, kite-like tilings (based on the rhombitrihexagonal dual lattice) and floret-like tilings (based on the snub hexagonal dual lattice) can also be accommodated. Tobacco ringspot virus, a pseudo $T = 3$ capsid composed of 60 protomers that are each made of three similar-sized but nonidentical jelly roll beta barrels[43], offers an example of a $T_r^D(1,0)$ tiling in which each kite-like tile represents the three domains of a protomer (Fig. 5d). As the three domains are not identical (cf. Fig. 2b & c in Johnson and Chandrasekar[43]), a triangle would not be an appropriate geometric description for this three-domain architecture. By contrast, we propose that a rhombitrihexagonal dual tiling is an adequate model. This hypothesis can be tested via its implications for the radius of the particle (cf. the equivalent argument for phage Basilisk and HSTV-2 in the Results section). In particular, the radius $R_T$ of Tobacco ringspot virus should be rescaled with respect to that of other members of the single jelly roll lineage such as Pariacoto virus (Fig. 4a), a $T^D(1,1)$ geometry with radius $R_P$, according to their respective lattice geometries as follows (see Supplementary Material for details):

$$\frac{R_T}{R} = \sqrt{\frac{\alpha_r}{3}},\tag{3}$$

with $\alpha_r$ as in Eq. 2. The average radii reported on the ViPER data base for each virus ($R_T = 15.4\,\text{nm}$ based on PDB 1A6C, and $R_P = 17.2\,\text{nm}$ based on PDB 1F8V, respectively)[12] imply a ratio of $\approx 0.90$. This is within 1% of the value of $\approx 0.91$ consistent with Tobacco ringspot virus being a $T_r^D(1,0)$ architecture, but

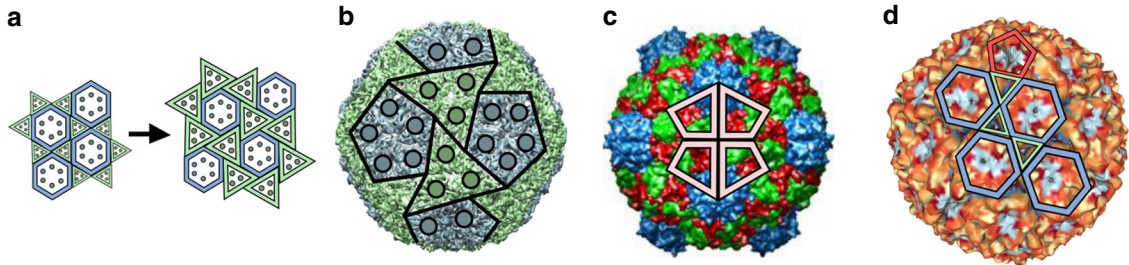

**Fig. 5** More general applications of the Archimedean lattice models in virology. **a** Rescaling of the triangular faces, with respect to the hexagonal ones, results in a gyrated version of the trihexagonal lattice in which proteins occupying different types of faces have identical footprints in the capsid surface. **b** The inner capsid of Pseudomonas phage phi6, a pseudo $T = 2$ structure formed from 120 chains, is an example of a gyrated $T_t(1, 0)$ lattice architecture. Its symmetry equivalent chains, shown in blue and green respectively based on RCSB PDB 4BTQ (ref. [42]), follow the layout of a surface architecture in which the surface areas of the triangular and pentagonal shapes occur in a ratio 3:5 (magenta), reflecting occupation by chains with comparable footprints on the capsid surface. **c** The kite tiling for Tobacco ringspot virus $\left(T_r^D(1, 0)\right)$, with tiles shown superimposed on a figure adapted from the ViPER data base (Tobacco ringspot virus: PDB-id 1a6c[43]). **d** A $T_t(2, 1)$ lattice, shown superimposed on a surface representation in rainbow colouring of RCSB PDB 2XYY[66], captures the outermost features of bacteriophage P22, with timers indicating the positions of the crevasses between the radially most distal features of the capsomers. Triangles also mark the positions of the trimer interactions between capsomers at a lower radial level

differs from the value $\sqrt{1/3} \approx 0.58$ expected if it was the same lattice type as Pariacoto virus.

Following Caspar and Klug's approach, we have used surface lattices in the Results section to indicate protein positions in the capsid surface. However, we note that when viewed at different radial levels, different types of lattice models may apply[44], revealing distinct aspects of capsid geometry. This is illustrated for bacteriophage P22. This virus is classified as a $T(2, 1) = 7$ Caspar-Klug geometry based on a hexagonal surface lattice. However, the organisation of its protruding structural features rather follows a trihexagonal lattice structure (Fig. 5c), consistent with the other architectures in the HK97 family discussed above. It is difficult to predict the additional lattices that can occur at different radial levels, unless their structures are coupled to the lattices describing the organisation of the capsid core discussed here. Such coupling could be modelled via affine extended symmetry groups[45–47] or 3D tilings[48], but this is beyond the scope of this paper. Interestingly, for the example of P22 the triangular positions correspond precisely to the trimer interactions between capsomers (cf. Fig. 5d in Thuman-Commike et al.[49]), suggesting that tiles may also have an interpretation in terms of interactions between capsomers. This had been observed previously in the context of Viral Tiling Theory for the cancer-causing Polyoma- and Papillomaviruses[45,50,51].

An intriguing observation is the lack of viral capsid examples adopting the regular rhombitrihexagonal and snub hexagonal lattices. One explanation could be that the sampling of possible viral structures is still rather limiting compared to the diversity of the virosphere[5]. There could also be physical explanations for the absence of such lattices. For example, the rhombitrihexagonal lattice requires a square tile, which may not occur in capsids as perhaps this may result in high mechanical stress, making such capsids less competitive. This is a phenomenon observed previously when comparing computational models for different viral architectures[32,52,53]. Some of these less thermodynamically favourable capsids, however, have been observed among mutant viruses in vitro, like the snub cube capsid with octahedral symmetry formed by 24 capsomers in papilloma virus, instead of the regular capsid with icosahedral symmetry[51,52,54]. Thus, it is possible that some of the absent lattices could be observed in the future as byproducts of in vivo or in vitro assembly of viruses and their mutants.

These lattices might also be of interest in nanotechnology and biomedicine, and provide inspiration for the construction of novel man-made icosahedral architectures across different length

scales. This may include architectures akin to Buckminster Fuller Domes[55], as well as protein containers in bionanotechnology and medicine, where they are used for a diverse range of applications, including vaccination, gene/drug delivery, phage display, imaging, energy and data storage[56,57]. In particular, icosahedral protein nanostructures assembled from pentamers and trimers (Figs. 1b and 3a in Bale et al.[58]) correspond to the smallest element in the trihexagonal series, and nanocontainers organised according to the duals of the Archimedean snub cube, i.e. the dual of the sunb hexagonal lattice, have also recently been reported[59]. These particles can be constructed akin to the icosahedral particles in Fig. 1 above, by superimposing the surface of an octahedron onto the different Archimedean lattice types; details of their surface architectures are given in the Supplementary Material. It is possible that larger structures in our icosahedral series, and their octahedral couterparts, may also be constructed from similar protein building blocks.

The polyhedral layouts describing the quasiequivalent capsid structures in CK-theory also occur in other areas of science, for example, as blueprints for the atomic positions in the fullerenes in carbon chemistry[60]. Similarly, the families of icosahedral polyhedra classified here can be applied to other chemical, physical and biological systems, for example, fullereneynes in chemistry[61], bound states of wave interacting particles in physics[62], and the iron storing encapsulin in biology[4], that all show the hallmarks of the $T_t$ architectures. The conceptual framework for the classification of icosahedral and octahedral polyhedral layouts presented here is therefore of interest for a wide range of scientific disciplines beyond virology.

## Methods
The construction of the polyhedral models and their duals is described below.

**Construction of polyhedral designs**. Consider two lines intersecting at an angle of 60° at the centre of one of the hexagons in the hexagonal (sub)lattice of a given Archimedean lattice. Counting steps between midpoints of adjacent hexagons along these lines via the integer coordinates $h$ and $k$, then $(h, k)$ characterises the positions of other hexagons in the (sub)lattice with respect to the original one, i.e., $(0, 0)$. Using the line connecting the midpoints of these hexagons as the edge of an equilateral triangle of an icosahedral face (Supplementary Fig. 1), the position of the remainder of that surface is uniquely determined, and $(h, k)$ thus defines a planar embedding of an icosahedral surface into the Archimedean lattice (see examples in Fig. 2). The corresponding polyhedral shape in three dimensions is an icosahedron, obtained via identification of edges of the planar embedding. The numbers of pentagonal, hexagonal, triangular and square faces in the Archimedean lattice overlapping with this icosahedral surface for different values of $h$ and $k$ are provided in Supplementary Tables 1–4 for the hexagonal, trihexagonal, snub

hexagonal and rhombitrihexagonal lattice, respectively. In particular, an icosahedral face given by $(h, k) = (1, 0)$ contains either no additional face (hexagonal case), one triangle (trihexagonal case), four triangles (snub hexagonal case), or one triangle and a square (rhombitrihexagonal case), that each form the start of an infinite series of polyhedra.

**Construction of the dual lattices**. For each polyhedron in the above classification, we construct a dual polyhedron. For this, vertices are positioned at the centres of the polyhedral faces, and vertices associated with adjacent faces connected by straight lines. Since Archimedean lattices have a single type of vertex environment, these dual polyhedra each have a single type of face that corresponds to the fundamental domain of a Laves lattice. These faces are triangles, rhombs, florets and kites for the hexagonal, trihexagonal, snub hexagonal and rhombitrihexagonal lattice, respectively. Using again the planar embedding of an icosahedral surface into the associated Archimedean lattice, we determine the numbers of each such face for polyhedra characterised by $h$ and $k$ as above; their numbers are listed in Supplementary Table 5.

**Measurement of capsid surfaces**. All surface measurements were carried out with UCSF Chimera[63].

**Reporting summary**. Further information on research design is available in the Nature Research Reporting Summary linked to this article.

## Data availability
Data supporting the findings of this manuscript are available from the corresponding authors upon reasonable request. A reporting summary for this Article is available as a Supplementary Information file.

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

## Acknowledgements

We are very grateful to Prof. Peter Stockley (University of Leeds) for his valuable insights in applications of this work to structural biology, to Dr. Richard Bingham (University of York) for his help with the figures, and to James Mullinix for suggesting Tobacco ringspot virus as an example for one of the dual surface lattice architectures. Financial support via an EPSRC Established Career Fellowship (EP/R023204/1) and a Royal Society Wolfson Fellowship (RSWF/R1/180009) to R.T. and a Joint Investigator Award to R.T. and Prof. Peter Stockley (110145 & 110146) are gratefully acknowledged. Financial support via the University Grant Program at SDSU to AL is also gratefully acknowledged. Molecular analyses were performed with UCSF Chimera, developed by the Resource for Biocomputing, Visualization, and Informatics at the University of California, San Francisco, with support from NIH P41-GM103311.

## Author contributions

R.T. and A.L. jointly designed and carried out the research, as well as contributed to the writing of the manuscript.

## Competing interests

The authors declare no competing interests.
