## [Peer Review File · Nature Communications]

Reviewers' Comments:

Reviewer #1:

Remarks to the Author:

In this paper Twarock and Luque advance a very clever extension of the Casper/Klug/Watson/Crick theory for the construction of viral capsids as projections onto Archimedean lattices and their duals. The theoretical framework they lay out for the classification and construction of viral capsids is relevant, timely and a clear advance to this field. The paper is well laid out, although largely written for the mathematically inclined. I recommend the paper for publication as is.

Reviewer #2:

Remarks to the Author:

This paper contains a wonderful observation about the structure of viruses and the Caspar and Klug classification system for virus structures. While in retrospect it seems rather surprising nobody has previously made this observation, I am fairly sure that it is new, and it certainly deserves to be published in a journal where it will be widely read. I recommend publication.

There was only one part of the paper that was confusing for me. It took me awhile to figure out what was meant by the last number in notations like $Tt(4, 0) = 64/3$ or $Tr \approx 2.49$. This should be explained more clearly.

The other thing I might suggest is that in Figure 2 (or in a new figure), the authors give an example of an Archimedean lattice that has some hexagons in addition to 12 pentagons. This is demonstrated in Figures 3 and 4, but it might be good to show the abstract mathematical example as well as the biological realization.

Reviewer #3:

Remarks to the Author:

This paper presents an interesting analysis of novel protein capsid assembly types that might be possible beyond the widely known and appreciated systems of icosahedral quasiequivalence laid out decades ago by Caspar and Klug. The vast majority of known viral capsids fall cleanly in the Caspar and Klug model, but some do not, and these have been hard to describe and categorize. The existence of divergent cases motivates alternative geometric schemes. The present paper goes to this point.

The specific proposal of this paper is that there are 3 Archimedean nets and corresponding polyhedra that can explain divergent types of viral capsids. Some known viruses are offered as examples for the 3 new geometries proposed.

The critical problem with the present study is that only one of the three geometric types introduced here (the 6,3,6,3 trihexagonal net) makes sense as a framework for describing real molecular capsids. And correspondingly, the authors offer potential cases of real viruses that can be explained by the three proposed geometries, but only the examples of the first type (based on HK97 type capsids) stand up to scrutiny.

The essence is that, despite being geometrically interesting, only one-third of what the authors offer here is biologically meaningful, which is the claim. It's possible that the study could be rescued if the authors were to recognize what has gone awry here, but it would be a different and considerably more concise paper.

Technical:

The problem with what has been done boils down to an inappropriate abstraction from pure geometry (polyhedra) to real molecular structure. Archimedean polyhedra are defined to have edges of equal length. The regularity of the faces so produced, however, does not always translate in a meaningful way to molecular ideas of oligomeric protein components on the faces. The molecular interpretation breaks down whenever the (equal-length) edges of a given face are non-equivalent with respect to the kinds of faces they occur between. Stated another way, a fatal molecular problem arises in cases where the symmetry of some face is not obeyed by the overall net (or polyhedron). In those cases, the presence of specific interactions between the proteins cannot be sustained over the broader assembly, even in the quasi sense. Note that in the first type of net/polyhedron offered (the trihexagonal or Tt case), which does seem to be a legitimate framework for molecular systems, the local symmetry of the triangular face is also obeyed by the symmetry of subunits arranged around the triangle. The edges all connect a triangle and a hexagon in the net. In the 3D icosahedral structure some of the hexamers will be converted to pentamers, but approximate symmetry is retained around the triangle. Indeed, the HK97 capsid types described do look like valid examples of this system.

In the other 2 types of polyhedra offered, triangles have edges that connect different types of faces in the net, and do not sit at points of 3-fold symmetry. Squares present in one of the nets suffer the same problem. It is particularly notable in these polyhedron types that the biological examples offered *do not* have oligomeric units sitting on the triangular and square faces as should have been predicted from the models in their simplest interpretations. Simply put, biological examples of these polyhedral types simply don't exist, apparently. What the authors have done in these cases is to abandon the basic expectation that oligomeric units should sit on these faces. Instead they say that certain parts or domains of the main capsid subunits contribute to these faces. This really just reduces these illegitimate cases to the simpler underlying Caspar and Klug hexagonal nets, possibly with somewhat irregular shaped subunits. The drawing of triangle and square faces on the known viral capsids in these cases takes the liberty of imagination and doesn't make physical sense in any rigorous way; the triangles don't represent trimeric protein components, nor do the squares represent tetrameric protein components.

The critical problem with these two Archimedean types almost certainly must have occurred to the investigators. Instead of trying to force these two impossible polyhedra only real structures, the authors need to recognize that only one of their three polyhedra is a valid model for expanding upon the Caspar and Klug hexagonal net models.

Response to the reviewers' comments

We are very grateful to all reviewers for their positive assessment of our mathematical approach. We agree with Reviewer 2 that a more detailed explanation of the new geometries would be helpful, and have therefore added such explanations after Equation (2) together with a new supplementary figure illustrating the geometries of larger structures in the new series. We also agree with Reviewer 3 that the biological examples used to illustrate the applications in virology were not optimal in all cases. In particular, there was a mixture of examples in which tiles were used to indicate individual protein subunits, or groups of subunits. To keep with the convention in Caspar-Klug theory, we now only present cases where tiles represent groups of proteins sharing equal footprints on a polyhedral face, or the domains of an uncleaved polyprotein as for example in the pT architectures of picornaviruses. We believe that these changes (see Figures 3 to 5 and the text passages highlighted in green) have remedied those shortcomings in the revised version and have indeed made the paper stronger as a result.

In particular, we have made the following changes:

- (1) Former Figure 4 is now Figure 3 in the revised version, emphasizing the examples of the Basilisk and other viruses in the HK97 lineage. These viral architectures correspond to different architectures in the trihexagonal series and illustrate conservation of this geometric blueprint within a viral lineage.
- (2) The new Figure 4 is an adaptation of the former Figure 5. It now exclusively provides examples in which tiles represent groups of proteins with equal footprint on the polyhedral face, or the domains of a polyprotein: Pariacoto virus (a triangulation, the dual of the hexagonal lattice), Bacteriophage MS2 (a rhomb tiling, the dual of the trihexagonal lattice), and Tobacco Ringspot virus (a kite tiling, the dual of the rhombitrihexagonal lattice).
- (3) The new Figures 5a&b contain a strongly improved exposition of the applications of the lattice models to Pseudomonas phage phi6.
- (4) The new Figure 5c illustrates how lattice models may apply at different radial levels of a capsid.
- (5) The former Figure 6a has now been included as a supplementary figure, and the corresponding text has been integrated with the paragraph on dual lattices. The example in former Fig 6b has been removed from the manuscript.
- (6) As requested by Reviewer 2, Figure 2 now provides a more explicit description of the nomenclature and values associated with the different T -numbers. It also refers to a new supplementary figure that shows folded structures associated to larger elements in the new icosahedral lattice series, as well as additional explanations after Equation 2.

Please note that in the revised version at least one viral example has been provided for either the lattice itself (the trihexagonal case) or for its dual (the hexagonal, trihexagonal and rhombitrihexagonal case), with the exception of the snub hexagonal lattice. However, an octahedral protein container following the dual of this lattice has just been reported in a recent Nature Letter (see new Ref 53), illustrating the importance of the snub hexagonal lattice case in nanotechnology. We have added this reference in the discussion and have also included information on the corresponding octahedral polyhedra in the Supplementary Material because of their relevance in nanotechnology.

Reviewers' Comments:

Reviewer #2:

Remarks to the Author:

I am very happy with the revisions.

Reviewer #3:

Remarks to the Author:

This manuscript has been heavily revised from its original form, both by removal of problematic ideas and by addition of entirely new themes.

It was raised in the initial review that of the 3 lattice systems elaborated by the authors, 3636 or 't', 33336 or 's', and 3464 or 'r', only the 't' type was suitable as a description of capsid subunit arrangements entirely distinct from those put forward by Caspar and Klug theory.

The revised manuscript doesn't explicitly recognize the symmetry problem pointed out in review that makes only the 't' type Archimedes lattice suitable for a packing system with different kinds of subunits packed on different kinds of polyhedral faces. But the paper does now correctly only attempt to identify the 't' system among natural viruses, and as before the HK97 family of capsids looks like an interesting and newly described application of this kind of lattice description (Fig. 3).

It seems that the 's' and 'r' Archimedes lattice types are no longer claimed to represent packings of atypical numbers of subunits on the polyhedral faces of known viruses. But instead of abandoning those lattice types, the revised paper takes a dramatic turn to consider how the duals of these lattices might offer different kinds of insights. But the question of dual lattices is really an entirely different kind of proposition, having more to do with the shapes of subunits on surfaces and less to do with the actual subunit composition or stoichiometry. This is a notable difference compared to the Caspar and Klug theory. The power of that theory is the specific predictions it made about subunit number, irrespective of what shape one might ascribe to the subunits. In the simpler cases presented here, the dual lattices shown by the authors (Fig. 4) seem to be just different ways of describing the shapes of ordinary or nearly ordinary T=3 Caspar and Klug packings. That the authors are able to designate these as duals of different (666, 3636, and 3464) lattices serves to emphasize that the dual descriptions do not seem to be defined in any clear and objective way by the virus composition. The assignments are in large part subject to artistic interpretations of shape. For the larger dual systems described (Fig. 5), it's not clear at all how these would be objectively analyzed. In the end, the dual descriptions proposed are mainly different ways of looking at things that aren't necessarily different from standard quasi-equivalence. The dual systems don't seem to make any specific predictions (as Caspar and Klug theory does) that could be tested or used to confirm or refute the idea. The paper's new major theme on duals, effectively unrelated to what was initially the main idea of viral systems with different kinds of facets, doesn't end up clarifying any puzzles or making unanticipated, concrete predictions that would give it robustness and impact.

Response to Reviewer #3

Please find enclosed a revised version, in which the latest changes are shown in magenta, in addition to the previous revisions shown in green.

In answer to the points raised:

It was raised in the initial review that of the 3 lattice systems elaborated by the authors, 3636 or 't', 33336 or 's', and 3464 or 'r', only the 't' type was suitable as a description of capsid subunit arrangements entirely distinct from those put forward by Caspar and Klug theory.

We provide a discussion of why the regular 's' and 'r' type lattices may not have been observed in virology to date (l. 315 – 326), but argue that these lattices could be of relevance to applications in bionanotechnology and in wider areas of science (l. 327 – 347). For that reason, we have opted not to remove them from the classification provided.

The revised manuscript doesn't explicitly recognize the symmetry problem pointed out in review that makes only the 't' type Archimedes lattice suitable for a packing system with different kinds of subunits packed on different kinds of polyhedral faces.

We have addressed this as follows: We have now strictly applied the convention of Caspar-Klug theory that polyhedral faces must represent groups of proteins that correspond, by number, to the rotational symmetry of the tile (cf. l. 196-199). As two of the dual lattices respect this (the triangulations and rhomb tilings, which are the duals of the hexagonal and trihexagonal lattice, respectively), we have kept those examples in Fig. 4.

There are two ways in which this strict convention can be relaxed while providing predictive results.

- (1) It is possible to accommodate more proteins on a tile as long as their number is divisible by the rotational symmetry of the tile. In particular, for the rhomb tilings, two rather than one dimer could be accommodated per tile, resulting in protein stoichiometries of $120T(h,k)$ (l.207-216). The picobirnavirus capsid is an example of that (see new Figure S3a). It is also possible to accommodate three dimers per rhomb tile, resulting in protein stoichiometries of $180T(h,k)$ (l.216-226). The Zika virus capsid is an example of that (see Figure S3b).
- (2) For viruses formed from a polyprotein (such as the $pT=3$ geometry of Tobacco ringspot virus) it is possible that the capsid protein domains adopt similar, yet distinct, conformations. In that case, a symmetric tile is not an adequate geometric description. However, the kite tiles corresponding to the duals of the rhombitrihexagonal lattice can accommodate this. This hypothesis can be tested via implications of lattice geometry on capsid radius. See lines 291-314 in main text and the new section "Scaling of capsid radius with lattice type" in the Supplementary Materials (l. 577-593).

We believe that both cases are justified because there are explicit examples for each of them as mentioned above; we have therefore chosen to include them. However, in order to distinguish their geometries explicitly from those capsid architectures that strictly respect the Caspar-Klug convention, we address them separately to signal that they are a generalisation of the core principle underpinning Caspar-Klug theory.

But the paper does now correctly only attempt to identify the 't' system among natural viruses, and as before the HK97 family of capsids looks like an interesting and newly described application of this kind of lattice description (Fig. 3).

We have added further examples for the 't' type lattices (l. 174-183), further underlining their importance in virology.

It seems that the 's' and 'r' Archimedes lattice types are no longer claimed to represent packings of atypical numbers of subunits on the polyhedral faces of known viruses. But instead of abandoning those lattice types, the revised paper takes a dramatic turn to consider how the duals of these lattices might offer different kinds of insights. But the question of dual lattices is really an entirely different kind of proposition, having more to do with the shapes of subunits on surfaces and less to do with the actual subunit composition or stoichiometry.

As explained above, we have eliminated any arguments using duals to predict subunit shapes, but explicitly link tile shapes with subunit composition/stoichiometry as suggested by the referee. As explained above, the dual lattices result in predictive conclusions on that basis, and we have therefore opted to retain the duals in the revised version.

This is a notable difference compared to the Caspar and Klug theory. The power of that theory is the specific predictions it made about subunit number, irrespective of what shape one might ascribe to the subunits.

We agree, and have now adhered to this convention: symmetries of tiles correlate with protein numbers/stoichiometry.

In the simpler cases presented here, the dual lattices shown by the authors (Fig. 4) seem to be just different ways of describing the shapes of ordinary or nearly ordinary $T=3$ Caspar and Klug packings. That the authors are able to designate these as duals of different (666, 3636, and 3464) lattices serves to emphasize that the dual descriptions do not seem to be defined in any clear and objective way by the virus composition. The assignments are in large part subject to artistic interpretations of shape.

Any such ambiguity is eliminated. Tiles are used exclusively to predict protein numbers and positions, not capsid protein shapes, as this would indeed open up ambiguities due to artistic interpretation as the referee correctly points out.

For the larger dual systems described (Fig. 5), it's not clear at all how these would be objectively analyzed. In the end, the dual descriptions proposed are mainly different ways of looking at things that aren't necessarily different from standard quasi-equivalence.

Fig. 5 now contains:

a&b: An example of a gyrated trihexagonal lattice. This is important in order to show that the gyrated version does also occur in nature. Its interpretation in terms of stoichiometry is the same as for the non-gyrated lattice.

c: An illustration that the outermost features of a virus also conform to the new lattice architectures.

d: An example of a capsid in which domains of a polyprotein adopt similar, yet distinct, conformations. In that case, a symmetric tile is not an adequate geometric description as discussed in item (2) above, but the dual of the rhombitrihexagonal lattice can be used to correctly predict the capsid radius (see details above).

The dual systems don't seem to make any specific predictions (as Caspar and Klug theory does) that could be tested or used to confirm or refute the idea. The paper's new major theme on duals, effectively unrelated to what was initially the main idea of viral systems with different kinds of facets, doesn't end up clarifying any puzzles or making unanticipated, concrete predictions that would give it robustness and impact.

In the Results section, we have now focused on predictive statements, relating tiles with protein stoichiometry or capsid radius.

Reviewers' Comments:

Reviewer #3:

Remarks to the Author:

In making further revisions the authors have de-emphasized the alternate lattice types that don't allow proper symmetry in their faces, as recommended. And they have qualified some of their proposed ideas about how dual systems might be useful descriptions of complex viral capsids. The paper remains highly complex (a weakness in my view), but at this point that is perhaps outweighed by the notable positive additions that are brought out by a fuller explanation of the authors' $T(\text{sub } t)$ lattice system and tangible viral examples that it supports.

Response to Reviewer #3

The last remaining issue mentioned by the referee is that *“The paper remains highly complex (a weakness in my view), but at this point that is perhaps outweighed by the notable positive additions that are brought out by a fuller explanation of the authors’ T(sub t) lattice system and tangible viral examples that it supports.”*

Please see below for a list of changes we made in order to mitigate the complexity issue:

1. We have made the following additions to make the rationale of the applications in virology in the Results section clearer:
 - l.142: We explicitly state that our framework is applied to capsids formed from a major and minor capsid protein.
 - l. 206: We stress that we are only focussing on the trihexagonal lattice and its dual (i.e., rhomb tiling), as in this case the CK convention of interpreting tiles in terms of protein positions can be applied.
2. We appreciate that we have made a number of additions in the Discussion at different stages of revision that make it difficult to peruse. We have now reorganised them (and also swapped Fig. 5 c and d in order to match the new organisation) as follows: We have added an introductory paragraph for all (l. 273), tightened the text for each, and added a line at the start of each item in order to help the reader see swiftly what each generalisation is about, i.e.:
 - l. 277: non-edge-to-edge tilings
 - l. 291: relaxing the symmetry condition on tiles

We prefer not to remove any of these results as they are all backed up by viral examples, and we believe that these results are important.

All changes are highlighted in magenta.